# Improved Outcomes in Myelofibrosis after Allogeneic Stem-Cell Transplantation in the Era of Ruxolitinib Pretreatment and Intensified Conditioning Regimen—Single-Center Analysis

**DOI:** 10.3390/cancers16193257

**Published:** 2024-09-25

**Authors:** Sigrid Machherndl-Spandl, Sarah Hannouf, Alexander Nikoloudis, Otto Zach, Irene Strassl, Emine Kaynak, Gerald Webersinke, Christine Gruber-Rossipal, Holger Rumpold, Wolfgang Schimetta, Johannes Clausen, Veronika Buxhofer-Ausch

**Affiliations:** 1Department of Internal Medicine I: Hematology with Stem-Cell Transplantation, Hemostaseology and Medical Oncology, Ordensklinikum Linz—Elisabethinen, 4020 Linz, Austriaalexander.nikoloudis@ordensklinikum.at (A.N.); irene.strassl@ordensklinikum.at (I.S.); emine.kaynak@ordensklinikum.at (E.K.); holger.rumpold@ordensklinikum.at (H.R.); veronika.buxhofer-ausch@ordensklinikum.at (V.B.-A.); 2Medical Faculty, Johannes Kepler University, 4020 Linz, Austria; 3Laboratory for Molecular and Genetic Diagnostics, Ordensklinikum Linz, 4020 Linz, Austria; otto.zach@ordensklinikum.at (O.Z.); gerald.webersinke@ordensklinikum.at (G.W.); 4Institute of Clinical and Molecular Pathology, Ordensklinikum Linz, 4020 Linz, Austria; christine.gruber-rossipal@pathologieverbund.at; 5Department of Applied Systems Research and Statistics, Johannes Kepler University, 4040 Linz, Austria; wolfgang.schimetta@asoklif.at

**Keywords:** myelofibrosis, allogeneic stem-cell transplantation, conditioning therapy

## Abstract

**Simple Summary:**

Primary myelofibrosis and secondary myelofibrosis after polycythemia vera and essential thrombocythemia are sub-entities of myeloproliferative neoplasms with poor prognosis for long-term survival. Allogeneic stem-cell transplantation remains the only potentially curative therapy, but the risk of serious side effects and possible treatment-related mortality must be considered. In recent years, efforts have been made to optimize conditioning therapy, donor selection, and supportive therapy in order to improve the outcomes of stem-cell transplantation. We report on 36 patients with myelofibrosis who underwent stem-cell transplantation in our center. We compared the outcomes in earlier and more recent years and evaluated the influence of certain transplant- and patient-specific variables. Pretreatment with a JAK inhibitor, intensified conditioning, and the preferential use of haploidentical instead of mismatched unrelated donors for patients lacking an HLA-identical donor are most likely responsible for the improved outcome after allogeneic stem-cell transplantation in myelofibrosis in recent years.

**Abstract:**

(1) Background: Allogeneic hematopoietic stem-cell transplantation (allo-HSCT) is the only treatment with the potential for cure in patients with myelofibrosis (MF). However, the risk of graft rejection, which is particularly high in MF, and the risk of significant non-relapse mortality must be considered. (2) Methods: In this retrospective, single-center study, we compared allo-HSCT outcomes in 36 adult patients with MF transplanted at two-time intervals (2001–2015 versus 2016–2021). (3) Results: The estimated median overall survival was 48.9 months (95%CI 0.00–98.2) in the cohort transplanted before 2016 and not reached in the more recent years (*p* = 0.04) due to markedly lower non-relapse mortality (*p* = 0.02). The 3-year relapse incidence was low in both cohorts (11.1% and 12.5%, *p* > 0.99). When comparing only subgroups within the more recent cohort based on the presence or absence of total body irradiation (TBI) or the use of sequential regimens, OS and PFS were comparable. (4) Conclusion: Pretreatment with ruxolitinib, intensified conditioning, and the preferential use of haploidentical related instead of mismatched unrelated donors for patients lacking an HLA-identical donor are most likely responsible for the improved outcome after allo-HCT in MF in recent years.

## 1. Introduction

Philadelphia-negative myeloproliferative neoplasms (Ph-MPN) are clonal stem-cell disorders that are classified as a subgroup of myeloid and histiocytic/dendritic neoplasms according to the World Health Organization (WHO) 2022 classification [1]. In addition to an increased risk of thromboembolic events and disease-related constitutional symptoms, there is a significant risk of progression and transformation in acute leukemia [2,3]. Myelofibrosis (MF), which can occur as primary myelofibrosis (PMF) as well as secondarily after essential thrombocythemia (post-ET MF) or polycythemia vera (post-PV MF) [4], is the subentity with worst prognosis for long-term survival. Several risk scores have been developed and validated as predictors of disease course and overall survival (OS), above all, the International Prognostic Scoring System (IPSS) [5], the dynamic IPSS (DIPSS score) [6], and the DIPSS-plus score [7]. In addition, leukemia-free survival is influenced by molecular markers [8]. For secondary myelofibrosis (SMF) developing in PV and ET patients, the Myelofibrosis Secondary to PV and ET-Prognostic Model (MYSEC-PM), integrating clinical and molecular data, was established as a specific tool for decision-making [9]. New risk scores, in particular the mutation-enhanced IPSS70 (MIPSS70) [10] and the MIPSS70+, version 2.0 (karyotype-enhanced MIPSS70) [11], include molecular and cytogenetic information to predict overall survival (OS) and were developed specifically for transplant fit patients (<70 years). The Myelofibrosis Transplant Scoring System (MTSS) was developed with the specific aim of predicting outcomes after hematopoietic stem-cell transplantation (HSCT), showing 5-year survival rates in the training (and external validation) cohort of 90% (83%), 77% (64%), 50% (37%), and 34% (22%) for the low, intermediate, high and very high MTSS risk category, respectively, and corresponding 5-year non-relapse mortality rates (in the combined cohort) of 10%, 22%, 36%, and 57%, respectively [12]. Especially for candidate selection for allogeneic hematopoietic stem-cell transplantation, PMF risk models should be simultaneously assessed [13]. The European Bone Marrow Transplantation (EBMT)/European Leukemia Network (ELN) consensus recommends HSCT for patients with intermediate-2- and high-risk DIPSS scores or a high-risk MIPSS (MIPSS70 or MIPSS70+) and a low or intermediate risk MTSS score [14].

In myeloproliferative neoplasms (MPN), disease-inherent inflammation contributes to patients’ symptoms, disease progression to myelofibrosis or acute leukemia, and thrombotic complications [15]. Multiple research groups have shown that chronic inflammation is sustained not only by the malignant clone but also by the cells of the bone marrow microenvironment, such as mesenchymal stromal cells, endothelial cells, and osteoblasts, as well as non-clonal hematopoietic cells [16,17,18]. Pretreatment should address these processes before an allo-HSCT, as they may impact the outcome of the procedure.

Treatment options for high-risk, advanced PMF, and transformed MF are limited. Ruxolitinib, a JAK1/2 inhibitor that reduces inflammation-related symptoms and splenomegaly was the first drug approved in patients with myelofibrosis in many countries (in Austria in 2012). The substance does not suppress fibrosis or the burden of malignant clones but reduces symptoms and may improve life expectancy in very high-risk patients [19,20]. Long-term data from the COMFORT-I/II trials showed a 30% reduction of mortality in intermediate-2/high-risk patients with ruxolitinib vs. controls [21]. Recently, two additional JAK inhibitors, such as fedratinib and momelotinib became authorized. However, after a certain period, most patients experience resistance to the tyrosine kinase inhibitors (TKI) with subsequent disease progression and risk of transformation to blast phase (BP). For this reason, patients with MF should be referred to a transplant center for evaluation before the onset of resistant, advanced disease.

Allo-HSCT remains the only potentially curative therapy for PMF and post-PV or post-ET MF. Reports in the literature suggest a relatively low relapse rate after allo-HSCT for MF, even after reduced-intensity conditioning (RIC) transplants, consistent with an immunological mechanism—graft-versus-MF—that is effective in eliminating the malignant clone without myeloablative chemotherapy [22], making transplantation feasible even in the elderly population. Advances in risk-adapted conditioning therapy and progress in transplantation medicine have led to improved outcomes in stem-cell transplantation for myelofibrosis. Using a dose-reduced conditioning protocol, a large trial from EBMT demonstrated a 5-year disease-free survival (DFS) of 51% and an OS of 67% [23]. In addition, pre-transplant ruxolitinib was associated with a high transplantation rate in a prospective multicenter study [24]. Recently, guidelines of the EBMT/ELN international working group and recommendations of “how to treat” were published [14,25].

In this retrospective analysis, we compared the outcomes of patients with MF after allo-HSCT in earlier and more recent years and evaluated the influence of transplant- and patient-specific variables in a cohort of patients who underwent transplantation at an Austrian EBMT center, the Ordensklinikum Linz—Elisabethinen (EBMT-CIC 594).

## 2. Materials and Methods

This single-center retrospective study included 36 consecutive patients who underwent allo-HSCT at the Ordensklinikum Elisabethinen Linz between 2001 and 2021. The diagnosis of MF was made according to WHO criteria [26].

The indication for allo-HSCT was based on standardized risk stratification; in general, patients with an estimated median OS of less than 5 years were considered for transplantation. In all patients with available data, the latest valid risk scores were retrospectively calculated. Accordingly, an intermediate or high-risk DIPSS-plus score [7] or a high-risk MIPSS70+ score [11] was assessed in 26/29 patients for whom data were available for recalculation. Table 1 shows the characteristics of the patients at the time of transplantation. Of the 23 patients who received reduced-intensity conditioning (RIC), the conditioning regimen consisted of either fludarabine (total dose, 150 mg/m^2^) in combination with intravenous busulfan 6.4 mg/kg (“FB2”; in total 12 patients, one patient instead received 8 mg/kg oral busulfan and 1 patient fludarabine and TBI 4Gy), or a modified, “FLAMSA”-derived [27] sequential protocol applying an induction phase consisting of “FLAC” (fludarabine 30 mg/m^2^/day and ARA-C 2000 mg/m^2^/day for 4 days), followed 3 days later by a busulfan (6.4 mg/kg; 6 patients) or total body irradiation (TBI) (4Gy; 4 patients)--based RIC. Reduced-toxicity myeloablative conditioning (RTC, *n* = 13) consisted of either fludarabine (total dose, 150 mg/m^2^) in combination with intravenous busulfan 9.6 mg/kg (“FB3”, 11 patients) or busulfan/fludarabine-based RTC regimens augmented with thiotepa (*n* = 1) or TBI 2Gy (*n* = 1). Sequential RTC consisted of “FB3” preceded by a 4-day FLAC induction (6 patients). Anti–T-lymphocyte globulin (ATLG) (Grafalon, Neovii Biotech, Gräfelfing, Germany) was added at a median total dose of 25 mg/kg (range, 20–45 mg/kg) in 19 HSCT with matched or partially mismatched unrelated (MUD or MMUD) or matched related donors (MRD). A total of 13 of the RIC patients and 6 of the RTC patients received ATLG.

For further analysis of the extent of influence of pre-transplant ruxolitinib as well as different conditioning strategies on the outcome, we divided the cohort into two subgroups according to the timepoint of transplantation: cohort 1 between 2001 and 2015 (*n* = 15), cohort 2 between 2016 and 2021 (*n* = 21)

The study was approved by the local ethical committee at Johannes Kepler University Linz (EC-Votum 1090/2022) and was conducted in accordance with the Declaration of Helsinki and Good Clinical Practice guidelines.

Response evaluation after allo-HSCT included engraftment, transfusion independency, spleen response, reduction in allele burden of driver mutations, and reduction in reticulin fibrosis. Bone marrow biopsy was performed at 6 and 12 months after transplantation, while chimerism was assessed periodically from day 30 onwards. Post-transplant engraftment was defined as absolute neutrophil count (ANC) recovery, with a neutrophil count > 0.5 × 10^9^/L in three consecutive days and a platelet (PLT) count maintained > 20 × 10^9^/L without transfusion for more than 7 days. Primary graft failure was defined as failure to achieve adequate neutrophil and platelet counts at day 60, and secondary graft failure was diagnosed as a decline in counts after the achievement of sustained adequate counts with loss of graft (recipient chimerism). Acute Graft versus host disease (GvHD) was graded according to Glucksberg or modified Glucksberg criteria, respectively [28]. Chronic GVHD was classified according to the Seattle criteria [29] as either “limited” or “extensive” or as “mild”, “moderate”, or “severe” according to the National Institutes of Health (NIH) 2005 consensus criteria, respectively [30].

To evaluate driver and non-driver mutations, molecular profiling of patients was performed by next-generation sequencing (NGS) using a capture-based customized next-generation sequencing (NGS) panel including 39 genes.

The dynamic of the allele burden over time was assessed by quantitative real-time polymerase chain reaction (Q-PCR) in patients carrying a Janus Kinase 2 V617F mutation (JAK2V617F) (sensitivity 0.0001%) and by classical PCR in patients carrying a calreticulin (CALR) mutation (sensitivity of 1%); therefore, a JAK2 allele burden < 0.0001% or a CALR burden < 1% were classified as complete molecular response (CMR). Moreover, molecular response was defined as a 25% (responder I) or 50% (responder II) reduction in relation to the allelic burden from baseline (baseline value = 100%), respectively.

OS and PFS were visualized by Kaplan–Meier plots, cumulative incidences of relapse, and non-relapse mortality by a competing-risk approach.

### Statistics

All data of continuous variables were checked for normal distribution (test of normality: Kolmogorov–Smirnov with Lilliefors significance correction, type I error = 10%). Continuous variables with normally distributed data were compared between the cohorts (Tx until 31 December 2015 vs. Tx since 1 January 2016) and between subgroups (see below) by the t-test for independent samples. For comparisons of continuous variables without normally distributed data and of variables measured on ordinal scales, the exact Mann–Whitney U test was used. Dichotomous variables were compared by Fisher’s exact test, and the other categorical variables by the exact chi-square test. For the comparison of the occurrence of events depicted by Kaplan–Meier plots, the log-rank test was used (where appropriate—namely in the case of time to death, time to relapse/progressive disease, and time to non-relapse mortality—the maximum possible observation periods in cohort HSCT until 31 December 2015 were adjusted to those in cohort HSCT since 1 January 2016 according to a matched pair principle). Competing risks were compared according to Gray RJ 1988 (Gray RJ. A class of K-sample tests for comparing the cumulative incidence of a competing risk. Annals of Statistics 1988; 16: 1141–1154). Since a multivariate approach (logistic regression analyses) was not effective for identifying the substantial influence of covariates on dichotomous endpoints due to frequent dependencies and too low number of cases (high number of missing values), a univariate approach (subgroup comparisons—methodology see above) was chosen. The influence of covariates on the occurrence of events was investigated using Cox regression analyses. However, a relatively large number of variables had to be excluded in these analyses because of the problems just mentioned. Two-sided 95% confidence intervals (CI) for incidences of events and survival were calculated according to Clopper-Pearson. The type I error was not adjusted for multiple testing. Therefore, the results of inferential statistics are descriptive only. Statistical analyses were performed using the open-source R statistical software package, version 4.2.3 (The R Foundation for Statistical Computing, Vienna, Austria).

Unless otherwise mentioned, data of continuous variables in the text reflect medians and quartiles.

## 3. Results

### 3.1. Patients and Disease Characteristics at Timepoint of Transplantation—Entire Cohort

The registry included 36 patients with the first allo-HSCT. Median post-HSCT follow-up was 31.7 (7.2; 61.2) months, corresponding to 2.7 (0.6; 5.1) years. Patients’ median age at the time of transplantation was 60.1 (range 40.4–74.9) years. Two thirds of the patients (69.4%) were males. Detailed baseline characteristics at the timepoint of transplantation are depicted in Table 1. Diagnosis was made according to the WHO classification [26]. PMF was the most frequent diagnosis, followed by myelodysplastic syndrome (MDS)/MPN overlap and secondary MF. Most patients were in the chronic phase of their disease. The median interval from diagnosis to transplantation was 12.4 (6.7; 31.1) months. A JAK2V617F mutation was detected in 21 patients (58.3%), and 9 patients (25.0%) exhibited a CALR mutation. Four patients were diagnosed with a thrombopoietin receptor (myeloproliferative leukemia protein, MPL)-positive MPN, and 2 patients were triple negative (TN) regarding a driver mutation. NGS revealed three or more so-called non-driver mutations in one third of the tested patients (30.3%), while half of the patients (52.9%) exhibited one or more high molecular risk mutations (HMR) prior to transplantation. Pronounced splenomegaly (length diameter equal to or more than 22 cm) was seen in almost a quarter of patients (23.5%) prior to transplantation. Exactly 50% of all patients had received ruxolitinib prior to transplantation. Fifteen patients (41.7%) were dependent on red blood cell transfusions, while only 11% were platelet transfusion dependent. Most patients (89.7%) were classified as intermediate or high risk according to the DIPSS-plus score prior to transplantation, and 83.3% were categorized into the high or very high-risk group according to the MIPSS70+, version 2.0 score. A high HCT-CI score (≥3 points) was assigned to 22.2% of all patients.

### 3.2. Patients and Disease Characteristics at Timepoint of Transplantation—Comparison of the Two Subcollectives

Patients and disease characteristics were comparable between the 2 cohorts with only one clear difference regarding the usage of ruxolitinib prior to transplantation; most patients (76.2%) transplanted after 2015 had received ruxolitinib pretreatment, compared to only 13.3% in the earlier transplanted cohort (*p* < 0.001) (Table 1).

### 3.3. Transplant Characteristics—Entire Cohort and Comparison of the Two Subcollectives

Conditioning regimens were more intensive in the transplantations performed from 2016 onwards, with nearly half of the patients (47.6%) receiving myeloablative RTC, while 80.0% of patients transplanted before 2016 received RIC regimens (*p* < 0.001). Substantially more patients transplanted before 2016 had an HLA mismatched donor (MMUD 26.7%;) compared to none in the more recent cohort; haploidentical transplantations (all have received post-transplant cyclophosphamide, ptCy-based GVHD prophylaxis) were exclusively performed in the more recent cohort (*p* = 0.009). FLAC-based sequential conditioning or TBI-based conditioning regimens were used only in the more recent cohort and were applied in 16/21 (76.2%) and 8/21 (38.1%) of transplantations, respectively. Transplant details are shown in Table 2.

### 3.4. Transplant Outcome—Entire Cohort and Comparison of the Two Subcollectives

Hematological recovery for absolute neutrophil count (ANC) and platelets (PLT) occurred in almost all patients (94.4 and 88.9%) after a median of 18.0 (16.0; 26.5) and 21.0 (16.0; 35.0) days, respectively. Eighty-one percent of patients achieved red blood cell (RBC)- and 91% of patients achieved PLT transfusion independence at day 100. Spleen length reduction by 25% (compared to the timepoint of transplantation) was achieved in one third (36.4%) of evaluated patients after 6 months and in 50% of evaluated patients after 12 months, respectively (details shown in Table 3).

Molecular response, defined as a 50% reduction of allele burden, was achieved in 88.2% of patients after 6 and in 92.3% of patients after 12 months, respectively. In absolute numbers, there was a median −97.63% (−98.09; −95.91) reduction of allele burden after 12 months (Appendix A). A complete molecular response was achieved in 7/17 (41.2%) JAK2 mutant patients at a median of 3.3 (0.9; 4.1) months and in 7/7 (100%) CALR patients at a median of 0.7 (0.6; 0.7) months. A clinically significant (≥2 points) improvement of reticulin fibrosis was seen in 27.8% of patients after 6 and in 57.1% of patients after 12 months, respectively.

Acute GVHD grade 2–4 and grade 3–4 occurred in 69.4% of patients. Extensive (or moderate/severe according to NIH consensus) chronic GVHD (cGVHD) was recorded in 33.3% of patients. Relapse or progressive disease occurred in only 2 patients (5.6%) after 5.1 and 27.4 months, respectively. These two patients were later retransplanted from an alternative donor. Furthermore, one patient with graft failure underwent a second transplantation with another graft rejection and was retransplanted a third time with finally successful engraftment.

The cumulative incidence of relapse was not different between the two cohorts (*p* = 0.83). The adjusted non-relapse death rate was clearly different, with 10 (66.66%) deaths in the earlier and only 3 (14.29%) deaths in the later transplanted group (*p* = 0.03). Causes of death were acute GvHD in 6 cases, chronic GvHD with pneumonia in one case, infection (2 cases of invasive fungal infection and 1 case of bacterial septicemia), relapse (one case), accident (one case) and one case of second primary malignancy (see Appendix A).

The estimated median overall survival was 48.9 months (95%CI 0.00; 98.2) in the cohort transplanted before 2016, whereas in the more recent period, it was not reached (*p* = 0.04). The cumulative incidence of non-relapse mortality was clearly different (*p* = 0.02); see Figure 1d. The estimated median progression-free survival (PFS) was 27.4 (95%CI 0.0; 62.2) months in the cohort transplanted before 2016, whereas in the more recent period, it was not reached (*p* = 0.02).

The calculated survival rate at 3 years post-HSCT of the whole cohort was 61.5% (95%CI 40.6–79.7%), the 3-year PFS 57.7% (95%CI 36.9–76.6%), the cumulative incidence of relapse (CIR) 11.8% (95%CI 1.5–36.4%) and the cumulative incidence of non-relapse mortality (NRM) 34.6% (95%CI 17.2–55.7%). Comparing the cohorts from 2001–2015 and 2016–2021, the 3-year calculated OS was 53.3% (95%CI 26.6–78.7%), and 72.7% (95%CI 39.0–94.0%), the 3-year PFS was 46.7% (95%CI 21.3–73.4%) and 72.7% (CI 39.0–94.0%), the 3-year CIR 12.5% (95%CI 0.3–52.7%) and 11.1% (95%CI 0.3–48.3%) and the 3-year cumulative incidence of non-relapse mortality (NRM) 46.7% (95%CI 21.3–73.4%) and 18.2% (90%CI 2.3–51.8%), respectively.

Kaplan–Meier curves for OS, NRM, relapse rate, and PFS of the two cohorts are shown in Figure 1a–d, and the KM estimated probabilities are in Appendix A.

### 3.5. TBI or Sequential Conditioning—Subgroup Analysis of Outcomes

When comparing patients who received conditioning with TBI (*n* = 8) to patients who received conditioning without TBI (*n* = 28), no substantial difference was found in estimated OS (median not reached versus median 92.5 months, 95%CI 36.04; 149.01, *p* = 0.29) and PFS (median not reached vs. median 29.3 months; 95%CI 19.19; 134.10; *p* = 0.23). Moreover, by comparing sequential (*n* = 16) with nonsequential conditioning (*n* = 20), no substantial difference was found in OS (median not reached vs. median 27.4 months, 95%CI 22.94; 130.34, *p* = 0.30) and PFS (median not reached vs. median 39.7 months; 95%CI 0; 154.42; *p* = 0.22). Furthermore, when comparing TBI- with non-TBI-containing conditioning (*n* = 8 vs. 13) and sequential (*n* = 16) with nonsequential regimens (*n* = 5) in the cohort from 2016–2021 only, again, no marked difference in OS and PFS was demonstrated. In the TBI versus non-TBI-group, the 3-year OS was 100% (95%CI 29.2–100%), and 62.5% (95%CI 24.5–91.5%) (*p* = 0.49), and the PFS was 100% (95%CI 29.2–100%) and 62.5% (95%CI 24.5–91.5%) (*p* = 0.49), respectively, the 3-year cumulative incidence of relapse 0% (95%CI 0–70.8%) and 16.7% (95%CI 0.4–64.1%) (*p* > 0.99) and the 3-year cumulative incidence of NRM 0% (95%CI 0–70.8%) and 25.0% (95%CI 3.2–65.1%) (*p* > 0.99), respectively. Comparing the sequential and nonsequential conditioning, the 3-year OS was 71.4.0% (95%CI 29.0–96.3%) vs. 75.0% (95%CI 19.4–99.4%) (*p* > 0.99) and the 3-year PFS was 71.4% (95%CI 29.0–96.3%) vs. 75.0% (95%CI 19.4–99.4%) (*p* > 0.99), respectively, the 3-year cumulative incidence of relapse 16.6% (95%CI 0.4–61.1%) and 0% (95%CI 0–70.8%) (*p* > 0.99) and the 3-year cumulative incidence of NRM 14.3% (95%CI 0.4–57.9%) and 25.0% (95%CI 0.6–80.6%) (*p* > 0.99), respectively. (see Kaplan–Meier curves in Appendix A and the KM estimated probabilities Appendix A).

### 3.6. Multivariate Analysis

#### 3.6.1. Logistic Regression Analyses

In the univariate subgroup comparisons performed as a substitute for the unstable multivariate logistic regressions, only the presence of ≥3 non-driver mutations and chronic GVHD appeared statistically remarkable. Extensive cGVHD correlated with a more pronounced improvement of reticulin fibrosis (*p* = 0.01), while the presence of 3 or more non-driver mutations correlated with a better spleen response at 12 months after transplantation (*p* = 0.03). However, the low patient numbers with available test results must be taken into account. More detailed results are depicted in the Appendix A.

#### 3.6.2. Cox Regression Analyses

Recent transplantation (between 2016 and 2021) was clearly associated with a longer OS (*p* = 0.003), a lower NRM (*p* = 0.004), and a longer PFS (*p* = 0.001). None of the variables examined correlated with ANC or PLT engraftment. More detailed results of the multivariate analyses are depicted in the Appendix A.

## 4. Discussion

Allo-HSCT remains the only curative therapy for PMF and post-PV or post-ET MF. However, the option of allogeneic transplantation is limited by age and comorbidities, as the median age of MF patients at diagnosis is approximately 67 years [31]. Our data show that outcomes for patients transplanted in a more recent era have improved notably, with a 3-year PFS of 57.7% for the entire cohort and a 3-year PFS of nearly 72.7% for patients who received an allograft in 2016 or later. This compares favorably with published data. For example, Robin et al. reported in the ruxolitinib-study a one-year DFS of 55% of the whole cohort and 83%, 40%, and 34% after matched sibling donor transplantation, transplantation with an unrelated donor with a 10/10 or 9/10 HLA match, respectively [24]. We have only observed a few relapses (2/36 patients, 5.6%) after a median observation time of almost 32 months, while the reported incidence in larger trials and registry data was notably higher. Kröger et al. reported a 22% cumulative incidence of relapse at 3 years in a cohort of 103 patients [23,24].

NRM was excessively high at 46.7% in the group transplanted before 2016, compared with only 14.3% in the more recently transplanted patients, even though transplant-specific risk scores, MTSS and HCT-CI, were comparable between the two groups.

Causes of non-relapse death in the earlier cohort were predominantly GvHD (6 cases) and fungal infection (2 cases in the years 2001 and 2003). It should be noted that posaconazole and voriconazole were not available at that time, but liposomal amphotericin was used for prophylaxis. One death was attributed to a second primary malignancy, and one to a car accident. In the second cohort, 3 patients died from infection, and one death was attributed to aGVHD. Although the incidence of acute and chronic GvHD was comparable in the two cohorts, the outcome was improved in the cohort of patients allografted after 2016, of whom 7 patients were treated with ruxolitinib for steroid-refractory aGVHD and 4 patients for steroid-refractory- or dependent cGVHD, while only one patient of the early cohort was treated with ruxolitinib after a long course of his cGvHD. Better GvHD control also appears to have influenced the improved outcome in the newer cohort.

The proportion of matched related donors was quite comparable in the two cohorts, whereas in the more recent cohort, one third of the donors (7/21) were haploidentical in contrast to the earlier transplant group when haploidentical transplantation (haploHSCT) was not yet established at our center. Preliminary data on haploHSCT demonstrated its feasibility in patients without suitable HLA-matched donors in myelofibrosis [32]. Six of our 7 patients with haploHSCT were alive at database lock. Although the number of patients is small, these outcome data are very promising. In this context, it should be noted that in the earlier cohort, all four patients who had received a mismatched unrelated donor transplant did not receive ptCy as this procedure was not yet established for MMUD transplantation but instead received GvHD prophylaxis with CsA/MMF, as well as ATLG. Furthermore, in the later cohort, MMUD could be completely avoided using haploidentical donors instead.

Mortality, graft failure, and the risk of aGVHD have been repeatedly reported to be higher in MF patients who had received an unrelated donor transplant [23,33,34]. In 2014, Gupta et al. found different five-year survival probabilities after allo-HSCT depending on donor selection in 233 patients with MF; the group reported a 56% survival rate with a matched sibling donor, 48% with a fully matched, and 34% with a partially matched/mismatched unrelated donor, respectively [35].

More recently, the question of antihuman T-lymphocyte immunoglobulin (ATLG) has been addressed in the setting of HLA-matched sibling donors in a few larger studies. An analysis of the EBMT registry for myelofibrosis patients showed a decreased risk of aGVHD without increasing the risk of relapse [36]. However, in a phase 3 study, ATLG had no impact on the rate of cGVHD in contrast to the data shown for transplantation in acute leukemia before [37]. In our study, the prophylaxis for GvHD varied significantly due to the high proportion of haploidentical donors in the later transplanted cohort. Prior to 2016, ATLG was primarily used in MUD and MMUD transplantations. However, starting in 2016, ATLG was also included in the GvHD prophylaxis for MRD transplantations. Unfortunately, due to the small number of patients, no definite conclusions can be drawn regarding the impact of ATLG on the incidence of GvHD and outcome.

Whether pre-transplant treatment with ruxolitinib, which was given to 76% of patients transplanted in the later period compared to only two patients in the earlier cohort, had an impact on the outcome can only be speculated. The JAK ALLO study was recently conducted to test the hypothesis that administering a short course of ruxolitinib prior to transplantation could improve post-transplant outcomes for MF patients. The study found that the use of ruxolitinib, which has immunosuppressive properties, did not increase the risk of disease progression and did not compromise the graft-versus-myelofibrosis effect. The relapse rate in this study was very low at 3% at 12 months. In addition, short-term ruxolitinib therapy before transplantation was associated with a high likelihood of successful transplantation in patients with a donor [24]. Another study demonstrated that patients who achieved clinical improvement on ruxolitinib prior to allo-HSCT had a lower risk of relapse (8% versus 19%) and better 2-year event-free survival (69% versus 54%) than those who did not [38].

The most beneficial effect of ruxolitinib prior to alloSCT may be explained by the fact that patients enter the transplantation phase with a reduced inflammatory state, an improved general condition, and a reduced spleen size [15].

According to the EBMT/ELN recommendations, all patients who are candidates for allo-HSCT with splenomegaly greater than 5 cm below the left costal margin or splenomegaly-related symptoms, should receive a spleen-directed treatment, ideally with a JAK inhibitor [14].

In our study, spleen size at the time of allo-HSCT did not differ between the two patient cohorts (*p* = 0.417), with a massively enlarged spleen ≥22 cm in only about a quarter of all patients, the median spleen length at allo-HSCT being 18 cm in both groups. Given the strong linkage of ruxolitinib pretreatment with the more recent transplant cohort, a univariate comparison of “RUX yes vs. no” would likely reflect the impact of the transplant date. In view of the methodical issues associated with the overall low number of observations, as discussed in the methods section, we acknowledge that our study is not powered to answer this question. Rather, we may conclude that transplant outcomes have significantly improved in the more recent era, which is characterized by several methodical changes, including donor choice, GVHD prophylaxis (particularly post-transplant cyclophosphamide for mismatched donors), and intensified conditioning, besides ruxolitinib pretreatment, and additionally, although not addressed in this study, improved GVHD management, which also started incorporating ruxolitinib in the more recent era.

Notably, although the more recently transplanted patients were treated with a more intensive conditioning regimen, with half of them receiving myeloablative conditioning with reduced toxicity, compared to most patients (80%) in the earlier group who received RIC, the NRM was markedly lower in the more recently transplanted group. As a result, median OS (not reached versus median 48.9 months, *p* = 0.04) as well as 3-year OS (79.6% versus 53.3%, *p* = 0.04) were markedly improved. According to the results of a large retrospective study by the EBMT, which analyzed the outcome of 2224 patients with MF who had received MAC or RIC conditioning, MAC is recommended for younger and adequately fit individuals due to a trend toward fewer relapses and a suggested benefit of improved GVHD-free/relapse-free survival (GRFS) [33]. While this study indicated an increased risk of graft failure in patients who had received RIC, our study did not show a notable difference in terms of PLT and WBC engraftment between the earlier cohort, where most patients underwent allo-HSCT with RIC conditioning and the later cohort. A meta-analysis of 43 studies with 8739 patients published in 2021 showed no significant difference in OS, NRM, GVHD, or graft failure between patients treated with different conditioning regimens [39].

In the case of blast phase MF, retrospective studies have reported the use of intensive induction regimens similar to those used in acute myeloid leukemia, with an acceptable CR rate prior to allo-HSCT [40,41]. Based on these results, we introduced a sequential conditioning regimen of fludarabine and high-dose cytarabine without G-CSF (“FLAC”) to reduce disease burden and spleen volume even in more advanced but still chronic phase patients, followed by fludarabine and either busulfan or TBI conditioning with or without ATLG in recent years. The regimen was very well tolerated, with only one early death due to infection, one case of graft rejection, and one relapse in a patient with the blast phase of the disease at the time of allo-HSCT. Comparison of the sequential protocol versus standard conditioning regimens showed comparable OS, no difference in relapse rate, and a remarkable difference in NRM (17.7% and 52.6%, respectively, *p* = 0.04), favoring sequential conditioning. However, when analyzing only the patient population transplanted since 2016, no differences in OS at 3 years, 3-year PFS, and cumulative incidence of relapse and non-relapse mortality were demonstrated between sequential and standard conditioning. Since sequential conditioning was mainly used in more advanced stages of MF, it seems to compensate for the poorer prognosis. Recently, a retrospective analysis by the Chronic Malignancies Working Group of the EBMT comparing sequential, myeloablative, and reduced-intensity conditioning in patients with myelodysplastic syndromes with blast excess at the time of allo-HSCT showed no significant differences in OS and RFS [42]. However, to our knowledge, data comparing these regimens in myelofibrosis have not been published yet.

We then asked whether the inclusion of TBI in the conditioning protocol affected the outcome in our cohort. Several groups had demonstrated the feasibility of adding low-dose TBI (2000–4000 centigray, cGy) to busulfan and fludarabine conditioning to reduce graft rejection [43,44]. In our cohort, only one patient who experienced graft rejection after RIC conditioning had a second graft failure after conditioning therapy, including 2000 cGy TBI, and was finally successfully transplanted a third time (with treosulfan, fludarabine and ATLG).

Analysis of the TBI-based conditioning regimens in our more recent cohort showed a promising OS and PFS of 87.5% in the TBI group, with no relapse and a low NRM of 12.5% in the TBI cohort. However, when interpreting these results, the small number of patients with TBI must be taken into account.

In closing, we would like to state that the primary constraints of our retrospective study were the limited sample size and the heterogeneous nature of the disease and treatment characteristics.

## 5. Conclusions

In this retrospective study, we have demonstrated a marked improvement in the outcome of MF patients after allo-HSCT in recent times, mainly due to a lower NRM, even with intensification of the conditioning regimen and the use of a sequential protocol and/or TBI. In addition, the selection of haploidentical donors rather than MMUD appears to contribute to a better outcome. Pretreatment with a JAK inhibitor, which is now the recommended standard, may have had an important impact on the improved outcome due to its anti-inflammatory properties, including reduction in spleen size and improvement in patient condition. Our encouraging experience underscores the importance of efforts to further improve the management of transplantation in myelofibrosis, therefore enabling most transplant-eligible patients to be cured of an otherwise fatal disease.

## Figures and Tables

**Figure 1 cancers-16-03257-f001:**
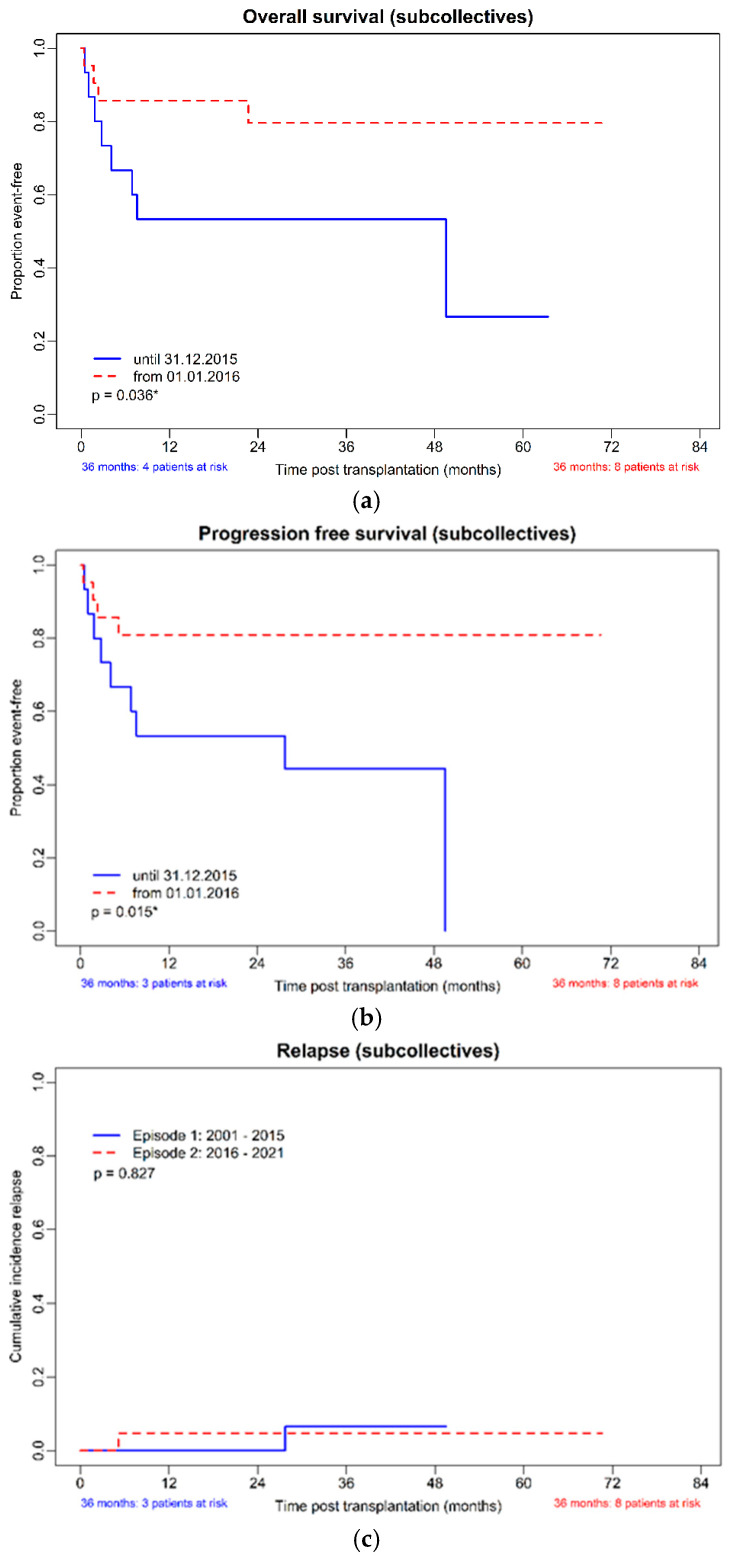
(**a**–**d**): OS, PFS, CIR, and NRM in the subcollectives (allo-HSCT until 31 December 2015, *n* = 15 and from 1 January 2016, *n* = 21). (**a**): Overall survival of the subcollectives. (**b**): Progression-free survival of the subcollectives. (**c**): Cumulative incidence of relapse of the subcollectives. (**d**): Non-relapse mortality of the subcollectives. * *p*-value with significance.

**Table 1 cancers-16-03257-t001:** Patient characteristics at the time of transplant.

**Variable**	**Entire Cohort**	**Subcollectives by Time of Transplant**		
	***n* = 36 ^#^**	**2001–2015** ***n* = 15**	**2016–2022** ***n* = 21**	***p*-Value**
Age, years, median (range)	60.06 (40.4; 74.9)	58.4 (43.6; 66.2)	62.7 (40.4; 74.9)	0.27
Gender, m (%)	25 (69.44)	10 (66.67)	15 (71.43)	>0.99
Interval initial diagnosis to transplantation, months, median (range)	12.4 (4.2; 394.9)	12.8 (4.6; 95.3)	12.0 (4.2; 394.9)	0.49
PMF or MDS/MPN with MF *n* (%)	21 (58.33)	7 (46.67)	14 (66.67)	0.23
secondary MF	11 (30.56)	7 (46.67)	4 (19.05)	
excess of blasts/blast crisis	4 (11.11)	1 (6.67)	3 (14.29)	
Chronic phase, *n* (%)	32 (88.89)	14 (93.33)	18 (85.71)	0.75
accelerated	2 (5.56)	0	2 (9.52)	
blast crisis	2 (5.56)	1 (6.67)	1 (4.76)	
Hb, mg/dL, median (quartiles)	9.40 (8.05; 11.15)	9.40 (7.40; 12.70)	9.40 (8.40; 11.10)	0.93
WBC, G/L, median (quartiles)	7.90 (4.75; 12.50)	5.50 (3.30; 18.60)	8.90 (5.20; 12.30)	0.62
PLT, G/L, median (quartiles)	119.50 (59.50; 224.50)	137.0 (86.0; 206.0)	94.0 (40.0; 282.0)	0.40
LDH, U/L, median (quartiles)	647.0 (444.5; 850.5)	536.0 (383.0; 737.0)	737.0 (558.0; 886.0)	0.10
Cytogenetics (Tx) unfavorable *, *n* (%), 26 pat.	6 (23.08)	2 (20.0)	4 (25)	0.77
Driver mutations				0.69
JAK2, *n* (%)	21 (58.33)	10 (66.67)	11 (52.38)	
CALR, *n* (%)	9 (25.00)	2 (13.33)	7 (33.33)	
MPL, *n* (%)	4 (11.11)	2 (13.33)	2 (9.52)	
TN **, *n* (%)	2 (5.56)	1 (6.67)	1 (4.76)	
Allele burden, % (quartiles), 27 pat.	31.0 (20.0; 53.0)	29.85 (18.0; 53.0)	33.0 (22.0; 52.0)	0.85
Non-driver mutations, ≥3, *n* (%), 33 pat.	10 (30.3)	5 (35.71)	5 (26.32)	0.70
HMR *** ≥1, *n* (%), 34 pat.	18 (52.94)	6 (42.86)	12 (60.0)	0.49
Spleen length, cm, median (quartiles), 34 pat.	18 (15.0; 21.0)	18 (15.0; 25.0)	18 (15.0; 21.0)	0.54
Spleen at Tx ≥ 22 cm, *n* (%), 34 pat.	8 (23.53)	5 (33.33)	3 (15.79)	0.42
Reticulin fibrosis, grade 0 or 1, 2 or 3, *n* (%), 29 pat.	5 (17.24)/24 (82.76)	3 (33.33)/6 (66.67)	2 (10.0)/18 (90.0)	0.29
Osteosclerosis, grade 0 or 1 vs. 2 or 3, *n* (%), 29 pat.	16 (55.2)/13 (44.8)	7 (77.8)/2 (22.2)	9 (45.0)/11 (55.0)	0.13
DIPSS, low/intermediate 1 or intermediate 2/high, *n* (%)	12 (33.33)/24 (66.67)	3 (20.0)/12 (80.0)	9 (42.86)/12 (57.14)	0.28
DIPSS-plus, low/intermediate 1 or intermediate 2/high, *n* (%), 29 pat.	3 (10.34)/26 (89.66)	0/12 (100)	3 (17.65)/14 (82.35)	0.25
MIPSS70 plus, low/intermediate or high/very high, *n* (%), 24 pat.	4 (16.67)/20 (83.33)	1 (10.0)/9 (90.0)	3 (21.43)/11 (78.57)	0.62
MTSS, low/intermediate or high/very high, *n* (%), 33 pat.	21 (63.64)/12 (36.36)	8 (57.14)/6 (42.86)	13 (68.42)/6 (31.58)	0.72
HCT-CI, ≥3, *n* (%), 33 pat.	8 (22.22)	3 (20.0)	5 (23.81)	>0.99
Ruxolitinib 3 months prior to Tx, *n* (%)	18 (50)	2 (13.33)	16 (76.19)	**<0.001**
RBC transfusion dependency prior Tx, *n* (%)	15 (41.67)	6 (40.0)	9 (42.86)	>0.99
PLT transfusion dependency prior Tx, *n* (%)	4 (11.11)	1 (6.67)	3 (14.29)	0.63

Hb Hemoglobin, PLT platelets, WBC white blood cells, LDH lactate dehydrogenase, RBC red blood cell. * Unfavorable: complex or 2 aberrations -7/7q-, i(17q), inv(3), -5/5q-, 12p-, 12q23. ** TN (triple negative). *** HMR (high molecular risk mutations: ASXL1, EZH2, SRSF2, IDH1/2, U2AF1Q157). # In case not otherwise stated. Significant *p*-values are shown in bold.

**Table 2 cancers-16-03257-t002:** Transplant characteristics, entire cohort, and comparison in the subcollectives.

Variable	Entire Cohort, *n* = 36	Subcollectives by Time of Transplant		
		2001–2015 *n* = 15	2016–2022 *n* = 21	*p*-Value
Sex mismatched female donor, *n* (%)	12 (33.33)	5 (33.33)	7 (33.33)	>0.99
CMV-positive recipient, *n* (%)	14 (38.89)	4 (26.67)	10 (47.62)	0.30
Reduced-toxicity/myeloablative conditioning (RTC)	13 (36.11)	3 (20.0)	10 (47.62)	**<0.001**
Reduced-intensity conditioning (RIC)	23 (63.89)	12 (80.0)	11 (52.38)	
TBI-based conditioning	8 (22.22)	0	8 (38.09)	
Sequential conditioning	17 (47.22)	1 (6.66)	16 (76.19)	
Donor type, MUD, *n* (%)	12 (33.33)	5 (33.33)	7 (33.33)	**0.009**
Donor type, MMUD, *n* (%)	4 (11.11)	4 (26.67)	0	
Donor type, MRD, *n* (%)	13 (36.11)	6 (40.0)	7 (33.33)	
Donor type, Haplo, *n* (%)	7 (19.4)	0	7 (33.33)	
Stem-cell source PB/ BM, *n* (%)	35 (97.22)/1 (2.78)	15(100)/0	20 (95.24)/1 (4.76)	>0.99
CD34 × 10^6^/kg BW, median (quartiles)	6.01 (4.86; 7.97)	5.04 (4.81; 7.32)	7.06 (5.0; 8.0)	0.26
GVHD prophylaxis, CSA/MMF, *n* (%)	26 (72.2)	13 (86.67)	13 (61.9)	**0.03**
GVHD prophylaxis, CSA/MTX, *n* (%)	3 (8.33)	2 (13.33)	1 (4.76)	
GVHD prophylaxis, ptCy/Tac/MMF, *n* (%)	7 (19.4)	0	7 (33.33)	

CMV, cytomegaly virus; TBI, total body irradiation; MUD, matched unrelated donor; MMUD, mismatched unrelated donor; haplo, haploidentical; PB, peripheral blood; BM, bone marrow; GvHD, graft versus host disease; CSA, cyclosporine; MMF, mycophenolate mofetil; MTX, methotrexate, ptCy, post-transplant cyclophosphamide; Tac, Tacrolimus. Significant *p*-values are shown in bold.

**Table 3 cancers-16-03257-t003:** Transplant outcome—entire cohort and comparison of the subcollectives.

Variable	Entire Cohort, *n* = 36 ^#^	Subcollectives at Time of Transplantation		
		2001–2015 *n* = 15	2016–2022 *n* = 21	*p*-Value
Follow-up, months (quartiles)	31.8 (7.2; 61.3)	10.3 (2.8; 36.8)	29.5 (14.5; 43.0)	0.14
ANC > 0.5 G/L, *n* (%)	34 (94.44)	14 (93.33)	20 (95.24)	>0.99
PLT > 20 G/L, *n* (%)	32 (88.89)	13 (86.67)	19 (90.48)	>0.99
Time to ANC > 0.5 G/L, days, quartiles	18.0 (16.0; 26.5)	17.0 (15.0; 30.0)	19.0 (16.0; 22.0)	0.97
Time to PLT > 20 G/L, days (quartiles), 35 pat.	21.0 (16.0; 35.0)	21.5 (17.0; 31.0)	21.0 (16.0; 46.0)	0.99
PLT transfusion independency day 100, *n* (%), 34 pat.	31 (91.18)	12 (85.71)	19.0 (95.0)	0.56
RBC transfusion independency day 100, *n* (%)	29 (80.56)	11 (73.33)	18 (85.71)	0.42
Delta Spleen length (6 mo-Tx, Tx = 100%),% (quartiles), 22 pat.	−17.95 (−27.48; −11.72)	−17.95 (−22.27; −14.78)	−20.28 (−29.59; −11.39)	0.67
Delta Spleen length (12 mo-Tx, Tx = 100%), % (quartiles), 16 pat.	−25.48 (−31.44; −16.91)	−25.0 (−30.98; −21.6)	−27.32 (−31.91; −15.08)	0.54
Spleen length at 6 mo, decrease >25% or unchanged, *n* (%); 22 pat.	8 (36.36)/14 (63.64)	2 (20.0)/8 (80.0)	6 (50)/6 (50)	0.20
Spleen length at 12 mo, decrease >25% or unchanged, *n* (%); 16 pat.	8 (50.0)/8 (50.0)	3 (42.86)/4 (57.14)/0	5 (55.56)/4 (44.44)/0	>0.99
Delta Allele burden,%, 6 mo-Tx (Tx = 100%), median (quartiles), 17 pat.	−96.86 (−97.8; −90.32)	−97.47 (−98.32; −93.59)	−96.25 (−97.63; −88.48)	0.30
Delta Allele burden, %, 12 mo-Tx (Tx = 100%), median (quartiles), 13 pat.	−97.63 (−98.09; −95.91)	−98.09 (−98.55; −90.32)	−97.45 (−97.91; −95.91)	0.57
Allele Burden Responder I, 6 mo, *n* (%), 17 pat.	15 (88.24)	4 (100)	11 (84.62)	>0.99
Allele Burden Responder II, 6 mo, *n* (%), 17 pat.	15 (88.24)	4 (100)	11 (84.62)	>0.99
Allele Burden Responder I, 12 mo, *n* (%), 13 pat.	12 (92.31)	3 (100)	9 (90.0)	>0.99
Allele Burden Responder II, 12 mo, *n* (%), 13 pat.	12 (92.31)	3 (100)	9 (90.0)	>0.99
Reticulinfibrosis improvement at 6 mo (0 + 1 points /2 + 3 points) *, 18 pat.	13 (72.22)/5 (27.78)	1 (33.33)/2 (66.67)	12 (80.0)/3 (20.0)	0.17
Reticulinfibrosis improvement at 12 mo (0 + 1 points/ 2 + 3 points) *, 14 pat.	6 (42.86)/8 (57.14)	1 (50.0)/1 (50.0)	5 (41.67)/7 (58.33)	>0.99
aGVHD (2–4, 3–4), *n* (%)	25 (69.44)	11 (73.33)	14 (66.67)	0.73
cGVHD (no or limited vs. extensive), *n* (%)	24 (66.67)/12 (33.33)	10 (66.67)/5 (33.33)	14 (66.67)/7 (33.33)	0.81
Death, *n* (%)	15 (41.67)	11 (73.33)	4 (19.05)	0.07
Non- relapse Death, *n* (%)	13 (36.11)	10 (66.66)	3 (14.29)	0.03
Time Tx to non-relapse death, days (quartiles), 13 pat.	84.0 (51.0; 228.0)	165.0 (55.0; 1486.0)	51.0 (12.0; 70.0)	0.11
Time to Death, days (quartiles), 15 pat.	123.0 (51.0; 1486.0)	207.0 (55.0; 233.0)	60.5 (31.5; 375.5)	0.23
Relapse/ PD, *n* (%)	2 (5.56)	1 (6.67)	1 (4.76)	>0.99
Time to relapse/PD, days, median (quartiles), 3 pat.	493.5 (155.0; 832.0)	832.0	155.0	>0.99

ANC, absolute neutrophile count; PLT, platelets; RBC, red blood cell; aGvHD, acute graft versus host disease; cGvHD, chronic graft versus host disease; PD, progressive disease. # In case not otherwise stated. * reduction in reticulin fibrosis measured in points.

## Data Availability

The raw data supporting the conclusions of this article will be made available by the authors upon request.

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
