# Peer review of "Improved Outcomes in Myelofibrosis after Allogeneic Stem-Cell Transplantation in the Era of Ruxolitinib Pretreatment and Intensified Conditioning Regimen—Single-Center Analysis"

_cancers, 2024, doi:10.3390/cancers16193257_

Round 1

Reviewer 1 Report

Comments and Suggestions for Authors

1. Summary

Machherndl-Spandl et al. review the outcome of 35 consecutive patients with PMF transplanted at their center between 2001 and 2021. The authors find markedly improved outcomes for patients transplanted from 2016-2021. This finding was mainly due to greatly reduced NRM, especially GvHD and infections. In the whole cohort there are only two patients who relapsed, whereas NRM in the early cohort was 66%.   

Author Response

Comments and Suggestions for Authors

Machherndl-Spandl et al. review the outcome of 35 consecutive patients with PMF transplanted at their center between 2001 and 2021. The authors find markedly improved outcomes for patients transplanted from 2016-2021. This finding was mainly due to greatly reduced NRM, especially GvHD and infections. In the whole cohort there are only two patients who relapsed, whereas NRM in the early cohort was 66%.   

Response: Since reviewer 1 does not provide any specific comments, we have carefully revised the manuscript based on the comprehensive comments of reviewer 2 and 3; we hope that the corrections also meet the considerations of reviewer 1.

Reviewer 2 Report

Comments and Suggestions for Authors

“Improved outcome in myelofibrosis after allogeneic stem cell transplantation in the era of ruxolitinib pretreatment and intensified conditioning regimen – single center Analysis”

The manuscript by Machherndl-Spandl et al. analyzes retrospectively, a cohort of 36 consecutive patients with primary or secondary myelofibrosis who underwent allogeneic stem cell transplantation (HSCT) between 2001 and 2021 at their institution. The authors divided this cohort into 2 groups according to the date of transplantation: cohort 1 (2001-2015) and cohort 2 (2016-2021) with the aim to reflect the influence of Ruxolitinib prior to transplant.  The authors then presented different transplant-related outcomes using analyzes of subgroups. 

1.        In table 1, it seems that patients with MDS/MPN overlap were included to those with PMF. Would clarify why were these patients included to the group with PMF. MDS/MPN overlap can be considered a different entity.

2.        In table 1, there is a typo in DIPSS plus, low/intermediate or 1/intermediate 2/high.  

3.        Line 214 mentions: “83.3% were categorized into the high or very high-risk group according to the molecular risk model (MTSS)” however in Table 1, only 12 patients belong to this category: 12 (36.36%). The 83.3% belongs to the high or very high risk group of the MIPSS70 plus score according to table 1.                                  

4.        In table 2, it is hard to interpret which patients received what type of conditioning regimen. In the narrative (Line 118), it says that 20 patients received “reduced RIC” however in the table says that RIC was received by 23 patients. Similarly, in the text (line 124), it says that 16 patients received reduced-toxicity myeloablative conditioning (RTC) whereas in the table says 13. Would also recommend mentioning how many patients received each of the conditioning regimens (how many received FB2, FB3, etc) and how many received sequential and non sequential conditioning within each group (RIC, RTC) as it is hard to interpret from the table. It is not clear which patients in each group received ATLG.           

5.        In table 3, it is hard to understand how data is presented for reticulin fibrosis improvement at 6 and 12 months. Would clarify that the categories 0+1 or 2+3 mean a reduction in points.

6.        The variable “exitus” is described in Table 3, would recommend providing with a definition of it in the text. Seems to be related to mortality.

7.        Line 277: would recommend including specific data of cumulative incidence of non-relapse mortality.                    

8.        Line 339 mentions 3-year PFS of nearly 79.6% for patients who received an allograft in 2016 or later however Supplemental Table 1 mentions 81%.

9.        Line 369 mentions that all 4 patients who had received a mismatched unrelated donor transplant did not receive PTCy; would recommend to mention what GvHD prophylaxis they received.

10.  Line 397, would recommend to add what was the relapse rate in the cited study.

11.  Line 437 mentions that there was “a strong trend toward lower NRM (p=0.04)”. This seems to be a significant difference rather than a strong trend, would suggest mentioning the actual values to clarify this as they are not mentioned in the text or tables (only data from the later cohort).

12.  Would suggest changing the title to outcomes (plural).

13.  The manuscript’s simple summary and introduction would perhaps benefit from additional editing for language.  A few suggestions:

Line 16: Would use poor prognosis instead of worst.

Line 20: would put outcomes in plural

Line 47: Would change “in” for into acute leukemia

Line 48: Would change “as well as” for “or”

Line 50: Would delete “the” in “the worst prognosis”

Line 52: Would change “Above all” for “above all are”.

Line 60: Would change “transplant fit patient” for patients who are fit for transplant

Line 73: Would erase “the”

Line 78: Would change “the” for “this”

Line 80” Consider deleting “general”

Line 86: Would change authorized to “FDA approved” Could potentially add indications for new Jak inhibitors.

Line 98: For patients with myelofibrosis instead of myelofibrosis patients.

Line 99: A large trial from EBMT

Line 103: Would change to outcomes (plural)

Line 104: Would delete “certain”

Line 111: diagnosis of MF was made according to WHO criteria. Would delete respective valid

Line 112: Would change to: indication for allogeneic HSCT was based on standardized risk stratification

Line 151: Delete S in health

Line 341: Would change to “a” instead of an. Would delete “so called”

Line 360 Would use “was” instead of “could”

Line 377 Would change to fully-matched instead of well-matched

Author Response

Review 2:

Comment 1.  In table 1, it seems that patients with MDS/MPN overlap were included to those with PMF. Would clarify why were these patients included to the group with PMF. MDS/MPN overlap can be considered a different entity.

Response: We included 19 patients with primary myelofibrosis and also 2 patients with MDS/MPN overlap with myelofibrosis grade 2 at diagnosis. The reason for the inclusion of patients with MDS/MPN overlap syndrome was the comparable management and prognosis of these patients.

Comment 2. In table 1, there is a typo in DIPSS plus, low/intermediate or 1/intermediate 2/high.  

Response: We’ve corrected the mistake.

Comment 3. Line 214 mentions: “83.3% were categorized into the high or very high-risk group according to the molecular risk model (MTSS)” however in Table 1, only 12 patients belong to this category: 12 (36.36%). The 83.3% belongs to the high or very high risk group of the MIPSS70 plus score according to table 1. 

Response: We corrected the text and replaced the MTSS by the MIPSS70+, version 2.0 score                                 

Comment 4. In table 2, it is hard to interpret which patients received what type of conditioning regimen. In the narrative (Line 118), it says that 20 patients received “reduced RIC” however in the table says that RIC was received by 23 patients. Similarly, in the text (line 124), it says that 16 patients received reduced-toxicity myeloablative conditioning (RTC) whereas in the table says 13. Would also recommend mentioning how many patients received each of the conditioning regimens (how many received FB2, FB3, etc) and how many received sequential and non sequential conditioning within each group (RIC, RTC) as it is hard to interpret from the table. It is not clear which patients in each group received ATLG.       

Response: 13 patients received MAC/RTC, 6 of them sequential conditioning with FLAC-FB3 +- ATLG, 1 Cy-FB3, 1 Thio-FB2, 1 FB3 + TBI 2Gy and 4 FB3 +- ATLG; ATLG Patients in total: 6 in the MAC/RTC group

23 Patients with RIC: 10 received sequential conditioning with FLAC (6 of them combined with TBI 4 Gy, 4 of them followed by FB2), 12 patients had FB2 +- ATLG and 1 patient Flu-TBI 4Gy; 13 of the RIC patients had ATLG

We corrected the text and expanded the information on the type of conditioning.

Comment 5. In table 3, it is hard to understand how data is presented for reticulin fibrosis improvement at 6 and 12 months. Would clarify that the categories 0+1 or 2+3 mean a reduction in points.

Response: An explanation in the table labeling was added.

Comment 6. The variable “exitus” is described in Table 3, would recommend providing with a definition of it in the text. Seems to be related to mortality.

We replaced the terminus exitus by death.

Comment 7. Line 277: would recommend including specific data of cumulative incidence of non-relapse mortality.   

 Response: A reference to figure 1d was added.      

Comment 8. Line 339 mentions 3-year PFS of nearly 79.6% for patients who received an allograft in 2016 or later however Supplemental Table 1 mentions 81%.

Response: We have corrected the numbers. In accordance to our statistician, we have used only the calculated survival rates in the main text, Kaplan Maier plot derived survival rates are mentioned in addition in the supplementary (in order to satisfy readers who are rather used to this kind of data presentation) .

 In principle, only cases that have had a chance of being observed for X years can be included in the calculation of an X-year survival rate. If this is the case with a KMP analysis, the results are identical (CI: de facto identical) to those from a cross-tabulation analysis. Otherwise, bias arises that can also become apparent through implausible results, e.g. higher survival rate for PFS than for OS - see extract from our analyses:

Comment 9. Line 369 mentions that all 4 patients who had received a mismatched unrelated donor transplant did not receive PTCy; would recommend to mention what GvHD prophylaxis they received.

Response: The 4 patients received CsA/MMF, as well as ATLG  for GvHD prophylaxis. This information was now included in the manuscript.

Comment 10.  Line 397, would recommend to add what was the relapse rate in the cited study.

Response: The relapse rate at 12 months was 3% (One-year DFS after transplantation was 55% and non-relapse-mortality was 42% at 12 months). This information was now included in the manuscript.

Comment 11.  Line 437 mentions that there was “a strong trend toward lower NRM (p=0.04)”. This seems to be a significant difference rather than a strong trend, would suggest mentioning the actual values to clarify this as they are not mentioned in the text or tables (only data from the later cohort).

Response: The following numbers were added in the manuscript for clarification and changes to the text made:… and a remarkable difference in NRM (17.7% and 52.6%, respectively, p=0.04).

The study did not have a confirmatory but an exploratory approach, the type I error was not adjusted for multiple testing. Therefore, p-values ​​<0.05 cannot be interpreted as significant, but must be classified as tendencies that are worth pursuing further (see also last section in the statistical description)

Comment 12.  Would suggest changing the title to outcomes (plural).

Response: We changed it to plural.

Comment 13:

  1. The manuscript’s simple summary and introduction would perhaps benefit from additional editing for language.  A few suggestions:

 Line 16: Would use poor prognosis instead of worst.

Line 20: would put outcomes in plural

Line 47: Would change “in” for into acute leukemia

Line 48: Would change “as well as” for “or”

Line 50: Would delete “the” in “the worst prognosis”

Line 52: Would change “Above all” for “above all are”.

Line 60: Would change “transplant fit patient” for patients who are fit for transplant

Line 73: Would erase “the”

Line 78: Would change “the” for “this”

Line 80” Consider deleting “general”

Line 86: Would change authorized to “FDA approved” Could potentially add indications for new Jak inhibitors.

Line 98: For patients with myelofibrosis instead of myelofibrosis patients.

Line 99: A large trial from EBMT

Line 103: Would change to outcomes (plural)

Line 104: Would delete “certain”

Line 111: diagnosis of MF was made according to WHO criteria. Would delete respective valid

Line 112: Would change to: indication for allogeneic HSCT was based on standardized risk stratification

Line 151: Delete S in health

Line 341: Would change to “a” instead of an. Would delete “so called”

Line 360 Would use “was” instead of “could”

Line 377 Would change to fully-matched instead of well-matched

Response:  Thank you for your helpful comments, we have corrected everything as suggested.

Reviewer 3 Report

Comments and Suggestions for Authors

This study retrospectively analyzed the results of allogeneic hematopoietic stem cell transplantation (allo-HSCT) for 36 patients with MF at a single center. Significant improvement was observed in the recent cohort even with intensification of the conditioning regimen and the use of a sequential protocol and/or TBI.

Major Comments
1.    The sample size of this manuscript was small. Thus, the results of this manuscript should be considered carefully. More cases are needed to identify the impact of certain transplant- and patients-specific variables (pre-transplant treatment, conditioning regimen, donor types, etc.) on outcomes of allo-HSCT for patients with MF.
2.    The characteristics for multivariate analysis should be provided in the main text. Furthermore, the information of univariate analysis is of importance and should be added in the article as a fact of low number of cases.

Minor Comments
1.    It’s better to use “MMURD” (lines 130 and 234) and “MMUD” (lines 242, 370, 371, 386, and 463) as a uniform term.
2.    I suggest that the causes of death could be delineated in the main text.
3.    It would be beneficial to reveal the outcomes of allo-HSCT in patients with pre-transplant ruxolitinib or not in the article

Comments on the Quality of English Language

less editing of English language required.

Author Response

Major Comments
Comment 1. The sample size of this manuscript was small. Thus, the results of this manuscript should be considered carefully. More cases are needed to identify the impact of certain transplant- and patients-specific variables (pre-transplant treatment, conditioning regimen, donor types, etc.) on outcomes of allo-HSCT for patients with MF.

Response: We certainly agree that a larger sample size would be desirable to enhance the reliability of the drawn conclusions. However, the study did not have a confirmatory but rather an exploratory approach, which was also taken into account in the statistical analysis and interpretation.  Because the type I error was not adjusted for multiple testing, p-values ​​<0.05 were not interpreted as significant, but were classified as tendencies that are worth pursuing further (see also last section in the statistical description).  We added a comment on the limitations of the study.

Comment 2. The characteristics for multivariate analysis should be provided in the main text. Furthermore, the information of univariate analysis is of importance and should be added in the article as a fact of low number of cases.

Response: 

Parameters included in the univariate and multivariate analysis were added in the supplementary:

Supplementary: Parameters included in the univariate and multivariate analysis

Gender (male / female); CMV status recipient (CMV positive / CMV negative);  Age at HSCT (years): <55 years, ³ 55 years;   CD34 x106/kg body weight recipient; Stem cell source (peripheral blood / bone marrow);  Conditioning regimen (MAC/RTC / RIC/NMA);  Donor type, dichotomous (MRD / other); GVHD prophylaxis, dichotomous (CSA/MMF / CSA/MTX or PTCy/Tac/MMF);  Osteosclerosis at HSCT, dichotomous (grade 0+1 / grade 2+3); Reticulin fibrosis bone marrow prior to HSCT, dichotomous (grade 0+1 / grade 2+3); Diagnosis at the time of diagnosis, dichotomous [PMF or MDS/MPN or sec. MF/ Excess of blasts (10-<20%)/ blast crisis (≥ 20%)]; State of disease at HSCT, dichotomous [chronic (blasts <10%) / accelerated (10−19% blasts) / secondary leukemia (≥20% blasts)]; RBC transfusion dependency prior to HSCT (≥ 2 RBC transfusions per month for at least three months prior to HSCT) (yes / no); PLT transfusion dependency prior to HSCT (≥ 1 PLT transfusion per month for at least three months prior to HSCT) (yes / no); Gender match (female donor to male recipient / other combination); aGVHD (2-4,3-4) (yes / no); cGVHD, dichotomous (no cGVHD+limited / extensive); Ruxolitinib ³3 months prior to HSCT (yes / no); Interval transplantation date to latest transplantation date (days); spleen size at transplantation (≥ 22cm)(yes/no); HCT-CI (³3 / <3);

Driver mutations at diagnosis, dichotomous (JAK2 + CALR / MPL + TN); Non driver mutations (≥3) at transplantation (yes / no); high risk mutations at transplantation  (ASXL1, SRSF2, EZH2, IDH1/IDH2 or U2AF1) (yes / no); DIPSS at HSCT, dichotomous (low risk + intermediate-1 risk / intermediate-2 risk + high risk); DIPSS plus at transplantation, dichotomous (low risk + intermediate-1 risk / intermediate-2 risk + high risk); MIPSS 70 plus at HSCT, dichotomous (low risk + intermediate risk / high risk + very high risk); MTSS at transplantation, dichotomous (low risk + intermediate risk / high risk + very high risk); Cytogenetics at diagnosis, dichotomous (normal or other / unfavorable)

Results of the univariate analysis were included in the results section

Multivariate analyses

Logistic regression analyses

In the univariate analysis subgroup comparisons performed as a substitute for the unstable multivariate logistic regressions only the presence of ≥3 non-driver mutations and chronic GVHD appeared statistically remarkable. Extensive cGVHD correlated with a more pronounced improvement of reticulin fibrosis (p= 0.01), while the presence of 3 or more non-driver mutations correlated with a better spleen response at 12 months after transplantation (p=0.03); However, the low patient numbers with available test results have to be taken into account. More detailed results are depicted in the supplementary.

Minor Comments
Comment 1.  It’s better to use “MMURD” (lines 130 and 234) and “MMUD” (lines 242, 370, 371, 386, and 463) as a uniform term.

Response: We corrected the text and used uniformly MMUD.

Comment 2.  I suggest that the causes of death could be delineated in the main text.

Response: We added the causes of death: Causes of death were acute GvHD in 6 cases, chronic GvHD with pneumonia in one case, infection (2 cases of invasive fungal infection and 1 case of bacterial septicemia), relapse (one case), accident (one case) and one case of second primary malignancy.

Comment 3.  It would be beneficial to reveal the outcomes of allo-HSCT in patients with pre-transplant ruxolitinib or not in the article.

Response:  Given the strong linkage of ruxolitinib pretreatment with the more recent transplant cohort, univariate comparison of “RUX yes vs no” would likely reflect the impact of the transplant date. In view of the methodical issues associated with the overall low number of observations, as discussed in the methods section, we acknowledge that our study is not powered to answer this question. Rather, we may conclude that transplant outcomes have significantly improved in the more recent era, which is characterized by several methodical changes, including donor choice, GVHD prophylaxis (particularly post-transplant cyclophosphamide for mismatched donors), and intensified conditioning, besides RUX pretreatment, and additionally, although not addressed in this study, improved GVHD management, which also started incorporating RUX in the more recent era.

Round 2

Reviewer 3 Report

Comments and Suggestions for Authors The authors have addressed my comments.